# The Genomics of Hairy Cell Leukaemia and Splenic Diffuse Red Pulp Lymphoma

**DOI:** 10.3390/cancers14030697

**Published:** 2022-01-29

**Authors:** David Oscier, Kostas Stamatopoulos, Amatta Mirandari, Jonathan Strefford

**Affiliations:** 1Department of Haematology, Royal Bournemouth and Christchurch NHS Trust, Bournemouth BH7 7DW, UK; 2Institute of Applied Biosciences, Centre for Research and Technology-Hellas, 57001 Thessaloniki, Greece; kostas.stamatopoulos@certh.gr; 3Cancer Genomics Group, Southampton General Hospital, Tremona Road, Southampton SO16 6YD, UK; a.mirandari@soton.ac.uk (A.M.); jcs@soton.ac.uk (J.S.)

**Keywords:** hairy cell leukaemia, HCLc, HCLv, SDRPL, *BRAFV600E*, Tbet

## Abstract

**Simple Summary:**

Hairy cell leukaemia is a rare chronic lymphoid malignancy with distinctive clinical and laboratory features which include an enlarged spleen, low blood counts, and infiltration of the spleen and bone marrow, with lymphocytes that have a villous or hairy cytoplasmic border. Historically it has been responsive to a range of treatment modalities including splenectomy, alpha interferon, and more recently chemotherapy, but none are curative. This review describes the chromosome abnormalities, genomic mutations, DNA methylation patterns, and immunoglobulin gene usage in this disease. We then discuss how the discovery of a specific mutation in a single gene (*BRAF*), present in almost all cases but not in hairy cell variant or splenic lymphoma with villous lymphocytes, two other splenic lymphomas with similar features, has provided new insights into its biology, a new diagnostic test, and a new therapeutic target.

**Abstract:**

Classical hairy cell leukaemia (HCLc), its variant form (HCLv), and splenic diffuse red pulp lymphoma (SDRPL) constitute a subset of relatively indolent B cell tumours, with low incidence rates of high-grade transformations, which primarily involve the spleen and bone marrow and are usually associated with circulating tumour cells characterised by villous or irregular cytoplasmic borders. The primary aim of this review is to summarise their cytogenetic, genomic, immunogenetic, and epigenetic features, with a particular focus on the clonal *BRAFV600E* mutation, present in most cases currently diagnosed with HCLc. We then reflect on their cell of origin and pathogenesis as well as present the clinical implications of improved biological understanding, extending from diagnosis to prognosis assessment and therapy response.

## 1. Introduction

The 2017 WHO classification of haematological malignancies recognises classical hairy cell leukaemia (HCLc) as a discrete entity and its variant form (HCLv) and splenic diffuse red pulp lymphoma (SDRPL) as provisional entities [1]. HCLc is a rare chronic lymphoproliferative disorder, with an incidence of 0.4/100,000. It is approximately four times more common in men than women and typically presents in middle age, with fatigue, infections, and abdominal discomfort due to splenomegaly. A complete blood count frequently shows cytopenia with almost universal monocytopenia, and a blood film usually reveals small numbers of medium-sized lymphoid cells with a ‘kidney-shaped nucleus’, infrequent nucleoli, weakly basophilic cytoplasm, and hairy projections. A modest lymphocytosis is seen in 5–10% of cases. Splenic histology shows diffuse infiltration of the red pulp with atrophy of the white pulp, a pattern also seen in HCLv and SDRPL, in contrast to the predominant white pulp involvement found in Splenic Marginal Zone Lymphoma (SMZL), which is the subject of a separate review in this series. Although originally believed to represent a tumour of haemopoietic progenitor reticuloendothelial cells [2], immunophenotyping identified HCLc as a tumour of mature B cells, typically expressing CD19, CD20, CD200, Tbet, PD1, and four markers of diagnostic value: CD11c and CD103, components of integrin receptors, and CD25 and CD123, components of interleukin receptors, of which at least three are expressed in all cases. Hairy cells do not express CD5 or CD27. Immunohistochemistry of bone marrow trephines additionally shows the expression of cyclin D1, CD72 (DBA-44), and Annexin A1, a specific marker for HCLc among B-cell malignancies, while bone marrow aspiration is usually unsuccessful due to the presence of reticulin fibrosis [3,4].

HCLv has an incidence of 0.04/100,000, a median age at presentation of 70 years, and a male-to-female ratio of 1.5–2. It was first described in 1980 in two patients with bulky splenomegaly, a marked leucocytosis with villous lymphocytes, and splenic histology showing red pulp involvement similar to that seen in HCLc, together with a number of distinctive features not found in HCLc. These include the absence of monocytopenia, larger tumour cells with prominent nucleoli, and bone marrow that is easy to aspirate due to absence or minimal marrow reticulin. Immunophenotyping shows the expression of CD19, CD20, and of the four archetypal HCLc markers, only CD11c and CD103 are commonly expressed, while CD25, CD123, and CD200 are negative or only weakly expressed. Immunohistochemistry shows the expression of CD72 but not annexin A1, and cyclin D1 is negative or weak [5,6].

In 2008, the term SDRPL was introduced to describe a further type of splenic lymphoma with circulating villous lymphocytes [7]. The frequency of SDRPL has not been established in the general population but represented 9% of splenic B-cell lymphomas seen in a 12-year period reviewed at the Spanish National Cancer Research Centre [8]. The clinical and laboratory features overlap those seen in HCLv, but the tumour cells generally lack a prominent nucleolus, and patients pursue a more indolent clinical course. Key demographic, morphological, phenotypic, and clinical differences between the three disorders are shown in Table 1.

## 2. Hairy Cell Leukaemia

### 2.1. BRAF V600E Mutations

The whole-exome sequencing of a single case of HCLc led to the discovery of a single somatic, point mutation in the DNA sequence of v-Raf murine sarcoma viral oncogene homolog B (*BRAF*), a kinase-encoding proto-oncogene. The same mutation was subsequently found in all 47 additional cases studied. The mutation replaces thymine (T) with adenine (A) in exon 15 of *BRAF* at position 1799 of the gene-coding sequence located in chromosome 7q34. In turn, this produces an amino acid change from valine (V) to glutamate (E) at position 600 (V600E) of the protein sequence, ultimately leading to aberrant activation of the *BRAF* oncogenic kinase and, thus, of the downstream MEK–ERK signalling pathway, such that ERK phosphorylation (pERK), detectable by immunohistochemistry, is a ubiquitous finding in *BRAF-V600E* positive HCLc [9,10]. The *BRAF-V600E* mutation in HCL is clonal and heterozygous, except in a minority of patients who lose the wild-type allele as a result of a concomitant 7q deletion [11]. Details of the RAS–RAF–MEK–ERK pathway and the *BRAF* protein with the site of the *BRAFV600E* mutation are shown in Figure 1 and Figure 2, respectively, and described in the accompanying legends.

*BRAF* mutations are found in a wide range of both solid and haematopoietic tumours, with a particularly high incidence in benign melanocytic nevi, malignant melanoma, papillary carcinoma of the thyroid [16], and the primary histiocytic disorders, Langerhans cell histiocytosis (LCH) and Erdheim–Chester disease (ECD) [17,18]. The clinical and biological consequences of *BRAF* mutations are highly variable and include the induction of a senescent phenotype, oncogenic transformation, and the emergence of secondary histiocytic sarcomas in a variety of acute or chronic, B or T cell, leukaemia, or lymphomas [19,20,21]. This variability may reflect the acquisition of additional genomic abnormalities such as the inactivation of cell cycle inhibitors, and the differentiation stage, transcriptomic and epigenetic features of the cell type in which the *BRAF* mutation arises [22,23].

#### 2.1.1. Haematopoietic Stem Cell Origin of *BRAFV600E* Mutation in HCLc

To identify the cell population from which the *BRAFV600E* mutation arises, immunophenotypically distinct CD34+, CD38− lineage-negative cells which encompass haemopoietic stem cells (HSCs) and their immediate multipotent progenitors [24,25], CD34+, CD38+ pro-B cells, myeloid progenitor cells, and HCLc cells were isolated with >97% purity from the bone marrow of 14 HCLc patients and age-matched controls [26].

HCLc patients were characterised by an expansion of HSCs and a marked decrease in the frequency of granulocyte-macrophage progenitor cells, consistent with the neutropenia and monocytopenia characteristics of HCLc. The *BRAFV600E* mutation was identified in the HSC, pro-B cell, and HCL cell populations, and quantitative sequencing analysis revealed a mean *BRAFV600E*-mutant allele frequency of 4.97% in the HSCs. Furthermore, in one patient who also had chronic lymphocytic leukaemia (CLL), the *BRAFV600E* mutation was present in both tumour cell populations, consistent with the mutation arising in a common precursor. To identify additional co-occurring genetic lesions that might cooperate with the *BRAFV600E* mutation to promote haematopoietic transformation, targeted mutational analysis was performed on HCL cells from three patients in whom the *BRAFV600E* mutation had been detected in HSCs. An additional *ARID1A* or *KMT2C* mutation was present in the leukemic cells but not the HSCs of 2/3 cases. HCLc patients treated with vemurafenib showed restoration of normal myelopoiesis, demonstrating that the impaired myeloid differentiation in HCL is dependent on mutant *BRAFV600E* signalling. This raises the question as to whether the clinical response to *BRAF* inhibitors may be mediated through their effects on mature leukemic cells, as well as through targeted inhibition of signalling and survival in mutant HSPCs.

Although both arise from HSCs, the co-existence of HCLc and LCH in the same patient has rarely been reported [27], possibly reflecting the different skewing of *BRAFV600E* mutant HPC differentiation in mouse models along lymphoid or myeloid pathways in HCLc and LCH, respectively. [26,28,29].

#### 2.1.2. Biological Consequences of the *BRAFV600E* Mutation in HCLc

The biology of HCLc reflects both cell-intrinsic factors as well as interactions with antigen(s), the extracellular matrix, the multiple cell types, and their secreted products present in the tissue microenvironment (TME) [30,31]. To ascertain the contribution of mutant *BRAF* to the HCLc phenotype, hairy cells from 26 patients were exposed in vitro to the specific *BRAF* inhibitors, vemurafenib or dabrafenib, or the MEK inhibitor trametinib. This resulted in the silencing of a gene expression signature which is specific to HCLc among B-cell tumours, with downregulation of genes including *CCND1*, *CD25*, and feedback inhibitors of ERK signalling such as members of the dual-specificity phosphatase (DSP) gene family. Additionally, *BRAF* or MEK inhibition caused loss of the hairy morphology and induced apoptosis which could be partially abrogated by co-culture with a bone marrow stromal cell line. *BRAF* and MEK inhibitors did not elicit any of the above-described biological effects in leukemic cells from four cases with HCLv, although the *MAP2K1* mutation status of these cases was not documented [32,33,34].

#### 2.1.3. Incidence of *BRAFV600E* in HCLc

The initial description of the *BRAFV600E* mutation in HCLc identified the mutation in all 48 patients tested [9], and several subsequent studies also found an incidence of 100% [11,35,36,37]. In contrast, other studies of patients reported having the typical clinical, morphological, and immunophenotypic features of HCLc have included a varying percentage of cases lacking the *BRAFV600E* mutation, with by far the highest incidence (21–25%) found among cases with relapsed/refractory disease [38,39]. Alternate methods of MAPK pathway activation have been discovered in some of these cases, including rare *BRAF* exon 11 mutations [40] and a single case with a t(7;14) (q34;q32) translocation resulting in an *IGH-BRAF* fusion [41]. The translocation juxtaposes the IGM switch region with exons 10–18 of BRAF, including the protein kinase domain, while removing the N-terminal auto-inhibitory Ras binding domain which spans exons 3–5. This results in ERK phosphorylation of tumour cells, indicating upregulation of MAPK signalling. Of potential clinical relevance, this patient would be expected to respond to MEK, but not *BRAFV600E*, mutation-specific inhibitors.

In a study of targeted gene sequencing in 20 HCLc cases, 2 lacked a *BRAF* mutation, of which 1 had a *MAP2K1* mutation, while no genomic abnormality was detected in the other case [42]. Among 53 cases with relapsed/refractory disease, *BRAF* was wild type in 11 (21%), including all 5 cases with clonotypic B-cell receptor immunoglobulin (BcR IG) utilising the *IGHV4-34* gene [38]. In a follow-up study of 27 HCLc cases, 7 were *IGHV4-34* positive, and activating *MAP2K1* mutations were identified in 5 *IGHV4-34*-positive but only 1 *IGHV4-34*-negative case [43].

### 2.2. Other Genomic Abnormalities

The most frequent cytogenetic and genomic abnormalities and immunogenetic features found in *BRAFV600E* mutated HCLc, *BRAF WT* HCLc, HCLv, and SDRPL are summarised in Table 2.

#### 2.2.1. Cytogenetic and DNA Copy Number Aberrations in HCL

Chromosome banding analysis (CBA) using a variety of B-cell mitogens identified clonal abnormalities in 70–80% of evaluable metaphases, but the nature and frequency of recurring abnormalities differed among studies. Abnormalities of chromosome 5, most commonly trisomy 5, or pericentric inversions and interstitial deletions involving band 5q13 were the most frequent abnormality in one study, detected in 12/30 cases [44]. Subsequent studies employing comparative genomic hybridisation also showed a varying incidence of copy number abnormalities (CNAs) but confirmed recurrent gains of 5q13-q31 and loss of 7q [45,46,47,48].

Two deep-targeted sequencing studies have enabled copy number analysis of regions sequenced by the panels. Among 53 *BRAFV600E* mutated cases of whom 22 were treatment naïve, recurrent abnormalities included deletions of 7q and of 13q14.3, encompassing *RB1* and the miR-15a and miR-16-1 microRNA cluster at 13q [11]. A second study of 20 cases sampled at diagnosis found loss of *MAPK15* in 7 (35%) of patients. The *MAPK15* gene, located on chromosome 8, encodes extracellular regulated kinase 8 (ERK8), a member of the MAPK family. The presence of a *MAPK15* CNA had no impact on treatment-free survival (TFS) and overall survival (OS), but progression-free survival (PFS) was significantly longer in cases with a *MAPK15* deletion [42].

#### 2.2.2. Somatic Genomic Mutations

Current information on the nature and incidence of somatic mutations other than *BRAF* in HCLc is based on limited data—namely, whole-exome sequencing (WES) of 9 cases [9,49,50] and targeted sequencing using a large panel of cancer-related genes in 73 cases [11,26,42], together with targeted sequencing of specific genes: *CDKN1B*, *MAP2K1*, and *KLF2*. Mutations in these genes are described in more detail below. Recurring low-frequency mutations also involve chromatin modifiers, discussed in the next section on epigenetic abnormalities and genes involved in Notch signalling (*NOTCH1* and *NOTCH2*), and DNA repair (*RAD50*).

CDKN1B

*CDKN1B* maps to 12p13 and encodes p27Kip1(p27), an intrinsically unstructured protein which regulates the transition from the G1 to the S phase of the cell cycle (Figure 3) and also has CDK-independent functions [51].

Low expression of p27 was demonstrated in all 58 cases of HCL studied and was associated with post-transcriptional downregulation, although the precise cause was not ascertained [53]. Subsequent studies in melanoma revealed a direct role for the *BRAFV600E* mutation: Expression of mutant *BRAF* was sufficient to upregulate cyclin D1 and downregulate p27 in human melanocytes [54], while in melanoma cells, mutant B-RAF controls p27Kip1 expression via mRNA abundance and proteasomal degradation [55]. A further potential mechanism for low p27 in HCLc emerged from microarray expression profiling of HCLc, compared with normal and other malignant B cells, which identified overexpression of miR-221/miR-222c which negatively regulates the expression of p27 [56,57].

More recently mutations of *CDKN1B* were identified in 13 of 81 (16%) patients with HCLc. All harboured at least one *CDKN1B* nonsense or splice site variant, except for one case in which a missense mutation was identified. Three patients had more than one mutation. implying selective pressure to inactivate *CDKN1B*. Overall, 11/13 *CDKN1B* mutations had allele frequencies very similar to those of the *BRAF* mutant clone, suggesting that *CDKN1B* mutations are early lesions that may contribute to HCLc pathogenesis by impairing cell cycle control and/or circumventing oncogene-induced senescence. *CDKN1B* mutations did not impact treatment response to PNAs [49].

KLF2

The Krüppel-like factor 2 (*KLF2*) zinc-finger gene, located at chromosome 19p13.1, encodes a transcription factor widely expressed in haemopoietic, endothelial, and lung cells. In B cells, *KLF2* regulates the expression of genes involved in cell homing, NF-κB signalling, and cell cycle control. B-cell-specific Klf2-deficient mice show a dramatic increase in cells with a marginal zone-like phenotype [58,59]. The KLF2 protein comprises activating and inhibitory domains, two nuclear localisation sequences (NLSs), and three zinc finger motifs (ZnFs). *KLF2* mutations are present in 20–40% of SMZL cases [60,61] and in three studies were also found in 9/74 (12%) cases of HCLc cases [42,62,63]. Mutations may occur in the activation, inhibitory, zinc finger or nuclear localisation domains, are predominantly truncating or missense, and reduce the transcriptional activity of *KLF2*, partly by displacement from the nucleus if mutations involve the NLS (Figure 4).

### 2.3. Germline Variants

Familial HCLc exhibits similar clinical features to sporadic HCLc but is rare, with fewer than 20 families reported in the literature. Four multiplex HCLc pedigrees were recently screened for shared germline variants, conferring HCLc susceptibility. Although there was only limited overlap between the pedigrees on a variant or gene level, several functional pathways such as neutrophil-mediated immunity and G-protein-coupled receptor signalling were shared in 3/4 and MAPK and RAS signalling in 2/4 pedigrees, respectively [66].

### 2.4. Epigenetic Abnormalities

The epigenome comprises chemical modifications to DNA and DNA-associated proteins which modify gene expression mediated through DNA methylation, histone tail modifications, chromatin accessibility, and DNA architecture and are critical for cellular differentiation and response to environmental stimuli. Post-translational modifications (PTM) of histone proteins regulate the accessibility of DNA to transcription factors and DNA repair enzymes by a variety of mechanisms which include recruiting additional chromatin-modifying factors, reducing the positive charge of histones, and altering the positioning of nucleosomes [67]. Mutations of genes that encode chromatin modifiers may also target nonhistone proteins.

#### 2.4.1. Mutations in Chromatin Modifiers

Mutations in genes involved in transcriptional regulation were found in 26/74 (35%) of HCLc cases [11,26,42]. The most frequently mutated gene was the histone methyltransferase *KMT2C* (MLL3) in which loss-of-function mutations throughout the coding region were identified in 15% (8 of 53) of cases [11]. *KMT2C* is a member of the KMT2 gene family which promotes methylation of H3K4 at enhancers and super-enhancers and transcription of genes related to cell differentiation or tumour suppression [68]. Other recurring mutations involve *CREBBP* and *EP300*, two interacting histone acetylation genes, and *BRD4*, *CEBPA*, *RUNX1*, and *MED12*. The transcriptional and phenotypic consequences of these mutations are highly context dependent, and their functional consequences in HCLc are unknown.

#### 2.4.2. DNA Methylation Profile

DNA methylation profiling was analysed with the low-resolution Infinium HumanMethylation27 array in 11 cases of *BRAFV600E* mutated HCLc, together with cases of CLL, SMZL, and normal B-cell subsets. HCLc had a distinct global methylation profile which, nevertheless, was more closely related to SMZL than to CLL and to normal post germinal centre (GC) memory B cells and marginal zone B cells than to pre-GC and GC B cells. When probes inside or outside cytosine guanine dinucleotide islands (CGIs) were analysed separately, the CGI-only methylation profile clustered all HCLc samples in an independent branch, separately from post-GC B cells but together with two of seven SMZL cases.

An integrated analysis of the HCL methylation profile and the previously published gene expression profile showed an inverse correlation between gene expression and methylation, alluding to a role for DNA promoter methylation in the regulation of specific gene expression. Independent supervised analyses were then performed to compare HCL methylation with that of post-GC B cells, SMZL, and CLL. Differential methylation changes observed in HCLc that were also reflected in gene expression patterns were consistent with constitutive activation of the RAS–RAF–MEK–ERK pathway and also affected pathways involved in the homing, migration, and survival of HCL cells [69].

### 2.5. Immunogenetic Features

The analysis of immunoglobulin gene repertoires in B-cell malignancies has provided key insights into their ontogeny, including their cell(s) of origin, the role and nature of antigenic stimulation in tumour development and evolution; moreover, in some diseases, especially CLL, *IGHV* gene SHM status has prognostic and predictive value [70].

While the majority of cases of HCLc have mutated *IGHV* genes, 17–20% are unmutated using a 98% cut-off and <5% have completely unmutated *IGHV* genes, with 100% identity to the germline. Compared with the normal B-cell repertoire [71], there is biased usage of the *IGHV3-21*, *IGHV3-30*, *IGHV3-33*, and *IGHV4-34* genes, each found in 7–10% of cases, with preferential use of *IGHV3-30* and *IGHV4-34*, especially among the unmutated cases. Biased usage of *IGHD* genes has also been documented [72,73,74,75].

While kappa is the most frequently used immunoglobulin light chain in the normal B-cell repertoire and in other B-cell tumours, HCLc is associated with preferential use of lambda light chains, resulting in an inverted Igκ:Igλ ratio (0.7:1). The explanation for this is unclear but may derive from secondary IG light chain gene rearrangements as part of receptor editing, a physiological process leading to the pairing of the authentic heavy chain with a novel light chain as a means to alleviate intense autoreactivity.

While there is much less diversity within the light chain, compared with the heavy chain repertoire, there is evidence of biased usage in HCLc, as virtually all cases that express lambda light chains utilise the IGLJ3 gene. In addition, the variable lambda complementarity determining region 3 (VL CDR3) of lambda-expressing cases frequently share structural features, while restricted pairings exist between certain conserved lambda light chains and heavy chains encoded by the *IGHV3-21/30/33* genes [72].

All of the above features support a role for antigen selective pressure in tumour ontogeny. Moreover, the presence of intra-clonal diversification within the clonotypic IG genes indicates that ongoing SHM occurs post-transformation likely in a context of continuous interactions with antigen(s).

An unusual feature of HCLc is the expression of multiple *IGH* isotypes on the cell surface, documented in 40% to over 80% of cases. Single-cell analysis has confirmed that this phenomenon is attributable to the expression of multiple isotypes in individual cells rather than to clonal heterogeneity [76]. Heavy-chain isotype switching is mediated through class-switch DNA recombination (CSR) which occurs between two switch (S) regions located 5′ of each *IGHC* gene. The intervening DNA segments are extruded via a cohesion-driven process and form extrachromosomal DNA switch circles. Deleted circle transcripts are not seen in HCLc cases expressing multiple isotypes, suggesting an arrest of CSR prior to deletional switching but where multiple isotypes can still be generated. CSR requires the upregulation of AID, enhanced chromatin accessibility mediated by histone modifications, and upregulation of factors such as IL-4, TGFβ, or IFNγ whose transcription is dependent on microenvironmental stimuli which determine the choice of specific isotypes. No genomic differences have been reported between cases that express either a single or multiple H chain isotypes, and the cause of the aberrant CSR remains uncertain [77,78].

The expression of multiple CH isotypes, including IgM with IgG or IgA, has also been reported in HCLv and SDRPL, although it has not been demonstrated if they are expressed in single cells. If so, this might point to a microenvironmental factor.

A further anomalous immunogenetic feature of HCLc is an increased incidence of cells expressing both IG kappa and lambda light chains. Dual expression of IG K and L light chains is rare in health, documented in only 0.2–0.5% of B cells from five normal controls [79]. Immunophenotypic analysis of 105 HCLc cases identified 3 (2.86%) that co-expressed surface kappa/lambda in virtually all cells. The immunogenetic analysis identified an additional case with a functional IGK/IGL transcript that also expressed multiple *IGH* isotypes and a RAG1 transcript. An increased incidence of dual light chain expressing cells is also seen in SLE [80] and in mouse models of autoimmunity [81,82,83]. The functional consequences of dual kappa and lambda light-chain expression are still unknown. That notwithstanding, evidence exists that compromised allelic exclusion leading to dual kappa/lambda expression might allow autoreactive cells to avoid clonal deletion, a mechanism described by the term receptor dilution [81].

### 2.6. Biological Implications: Cell of Origin

The identity and behaviour of tumour cells are largely controlled by the activity of transcriptional programs which reflect the programs active in, or available to, their cell(s)of origin, and/or their dysregulation by either cell-intrinsic genomic/epigenetic factors or cell-extrinsic interactions with the TME [84,85]. The biological and clinical significance of these variables has been well documented among mature B-cell tumours such as diffuse large B-cell lymphoma (DLBCL), mantle cell lymphoma (MCL), and SMZL [86,87,88].

The stem cell origin of the clonal *BRAFV600E* mutation in HCLc, together with the induction of a lethal haematopoietic disorder with features of HCLc in *BRAFV600E* mice [26], are consistent with the role of *BRAFV600E* as an early/initiating event in hairy cell leukemogenesis. However, there remains uncertainty about the nature of the mature B-cell population(s) expanded in HCLc and whether additional genetic and/or epigenetic alterations or a suitable TME are required to give rise to mature HCL cells. Dysregulation of the G1 phase of the cell cycle is a common finding in HCLc, but as yet, no subsequent genomic final transforming event has been discovered.

Data relevant to determining the COO in HCLc include the following:The presence of mutated *IGHV* genes, with evidence for antigen selection, in the majority of cases and preferential use of the *IGHV4-34* gene in the minority of cases with low or no SHM;A gene expression profile and methylome more similar to that of CD27 positive memory and marginal zone B cells than to naïve or germinal centre B cells [32,69];A phenotype which includes expression of CD11c+, Tbet+, and PD1+ but not CD27 [89,90,91,92].

This phenotype also delineates a subset of normal B cells present in blood and splenic red pulp but rarely in lymph nodes. These cells also lack expression of CD21 and the chemokine receptors CD185 (CXCR5) and CD184 (CXCR4), reflecting their distribution within lymphoid organs. They frequently express sIgG, consistent with the role of Tbet in regulating antibody class switching to IgG1 or IgG3. Cells with a CD11c+ Tbet+ phenotype are found within B-cell populations variously described as age-associated B cells, atypical B cells, and double-negative B cells. CD11c+, Tbet+ B cell numbers increase with age and are expanded in conditions associated with chronic antigenic stimulation such as infections with human immunodeficiency virus and malaria, and in autoimmune diseases such as SLE. However, the CD11c+ Tbet+ phenotype does not, by itself, identify a distinct B-cell population, nor a specific B-cell lineage, and can be found in activated naïve B cells and in memory B cells believed to be generated through follicular or extra-follicular maturation pathways [93,94,95,96,97,98,99].

Immunogenetic, transcriptomic, and epigenetic analysis of B cells based either on the expression of CD11chi, Tbet, or a double-negative phenotype (CD19+ IgD- CD27- CXCR5-) shows significant differences from canonical CD11-ve Tbet -memory B cells [96,100,101]. Interestingly, the CD11c+ cohort is associated with enrichment of *IGHV4-34* gene usage [100]. Immunoglobulins encoded by the *IGHV4-34* gene display autoreactivity to the I/i antigens present on erythrocytes by virtue of a germline motif within the VH FR1 and additionally show cross-reactivity with other self and microbial antigens. Naïve B cells expressing *IGHV4-34* are often anergic, while this gene is largely excluded from switched memory and plasma cells. The persistence of this gene in a spleen-resident Tbet+ memory B-cell subset has been postulated to reflect either positive selection of B cells that may facilitate clearing of self-antigens or neutralisation of microbial antigens [102] or a defect in negative selection during GC transit [100].

It would be interesting to review the transcriptomic and methylation data in HCLc using normal splenic CD11c+ Tbet+ CD27- cells rather than CD27+ memory B cells as the comparator. However, currently, and as in CLL despite extensive studies [103], the COO of HCLc remains enigmatic.

### 2.7. Clinical Implications of Genetic Features

#### 2.7.1. Diagnosis

The initial description of *BRAFV600E* in HCLc failed to find the same mutation in 195 cases of other mature B-cell tumours including CLL, follicular lymphoma, DLBCL, and other splenic lymphomas [9]. However, subsequent screening of larger cohorts of CLL and myeloma for both *BRAFV600E* and other *BRAF* hotspot mutations has identified a low incidence of predominantly subclonal V600E and non *V600E BRAF* mutations, usually associated with a poorer outcome [104,105,106,107,108].

Whilst a confident diagnosis of HCLc can be made without knowledge of the *BRAF* mutation status, the specificity of the *BRAFV600E* mutation for HCLc among splenic lymphomas is valuable when there is diagnostic uncertainty, and its presence underpins the use of targeted inhibitors. If it emerges that widely available diagnostic criteria are unable to distinguish *BRAFV600E* HCLc from *BRAFWT* HCLc, this would provide an additional rationale for *BRAFV600E* mutation screening. Allele-specific PCR performed on blood or marrow aspirate samples has superseded less sensitive molecular techniques such as Sanger sequencing, pyrosequencing, or melting curve analysis [109]. Digital, droplet PCR has comparable specificity and superior sensitivity to QT–PCR and is a potential method for MRD analysis [110]. Immunohistochemistry (IHC) using a *BRAFV600E*-specific antibody is an alternative method suitable for bone marrow trephine or other tissue sections, with comparable sensitivity and specificity to allele-specific PCR. Next-generation sequencing (NGS) also has high sensitivity but, currently, also has higher costs and longer turnaround time, compared with allele-specific tests [111].

#### 2.7.2. Prognostic Significance of *IGHV* Gene Somatic Hypermutation Status

The clinical significance of *IGHV* gene SHM status in HCLc has been evaluated in two studies with discordant results. In a trial of single-agent cladribine in 58 previously untreated patients, all expressing annexin A1, failure to respond was observed in 5/6 patients with unmutated *IGHV* genes using a 98% cut-off value, only one of whom used *IGHV4-34*. Bulky splenomegaly, leucocytosis, and *TP53* abnormalities were present in four, three, and two of the five cases, respectively [73].

In a cohort of 62 patients with HCLc and 20 with HCLv diagnosed according to the WHO 2008 criteria [112], *IGHV4-34* was used in 6 (10%) of HCLc and 8 (40%) of HCLv cases, respectively, and was unmutated in all but 1 case, using a 98% cut-off value. A suboptimal response to first-line treatment with cladribine was seen in 4/6 *IGHV4-34* HCLc positive cases, compared with 4/56 *IGHV4-34* negative cases. A worse response was also seen in *IGHV4-34* positive HCLv cases, suggesting that outcome was more closely related to *IGHV4-34* status than to whether or not patients had HCL or HCLv. However, many of the HCLc cases were *BRAFV600E* negative [113].

#### 2.7.3. *BRAFV600E* as a Therapeutic Target

The purine nucleoside analogues (PNAs), pentostatin, and cladribine remain the current treatment of choice for first-line therapy of HCLc. However, PNAs may cause short-term myelosuppression, with an increased risk of infection and an increased risk of secondary malignancies, and approximately 50% of patients eventually relapse. Single-agent vemurafenib or dabrafenib resulted in high overall response rates without minimal/measurable residual disease (MRD) negativity in relapsed/refractory HCL, but the median relapse-free survival in responders was less than 1 year [114,115]. In contrast, a phase II study of vemurafenib plus rituximab achieved a CR rate of 87%, of whom 65% were MRD negative and with relapse-free survival of 85% at a median follow-up of 34 months [116].

#### 2.7.4. Genomic Abnormalities as Predictors of Drug Resistance

To PNAs

Targeted mutational and copy number analysis showed no difference in the pattern of genomic abnormalities between treatment naïve cases and those refractory to a PNA [11]. Serial samples from two HCL-c cases tested both at diagnosis and relapse post-PNA therapy, revealed two additional subclonal mutations of *BCOR* (BCORE1430X) and *XPO1* (XPO1E571K) in one case, while the second case remained genomically stable [42]. However, there is no clear evidence to suggest that genomic mutations confer resistance to PNAs in HCLc.

To BRAF Inhibitors

Of 13 evaluable HCLc cases treated with vemurafenib, 6 showed persistence of ERK phosphorylation in bone marrow cells, suggesting that, in at least some patients, the growth of HCL cells remains dependent on MEK–ERK signalling, likely reactivated through mechanisms bypassing *BRAF* inhibition by vemurafenib. In support, targeted sequencing of 300 genes performed in one patient who was refractory to vemurafenib showed two separate activating subclonal *KRAS* mutations at relapse [116].

A further case with vemurafenib resistance had heterozygous deletions of *BRAF*, *NF1*, *NF2*, and *TP53* and subclonal mutations in *CREBBP* and *IRS1* in a pretreatment sample. *NF1* and *NF2* encode tumour suppressors that have been experimentally implicated in RAF inhibitor resistance in epithelial cancer cells [117] and downregulation of either or both Nf1 or Nf2 in Ba/F3 cells stably expressing *BRAFV600E* conferred vemurafenib resistance in vitro [11].

Seven distinct activating mutations in *KRAS* and two mutations in *MAP2K1* were detected in the relapse sample of a patient resistant to a PNA and vemurafenib plus rituximab. Allele frequencies were consistent with the parallel, convergent evolution of multiple clones with *KRAS* mutations appearing before *MAP2K1* mutations. Treatment with MEK inhibitor cobimetinib in combination with vemurafenib resulted in significant clinical and haematological improvement, associated with suppression of mutant allele frequencies for *BRAF*, *KRAS*, and *MAP2K1* mutations and of ERK activity [118].

Elucidating the mechanisms of resistance to *BRAF* inhibitors in solid tumours, especially melanoma, is an area of intensive investigation. In addition to the selection of genomic mutations such as mutations of *RAS* or *MAP2K1*/*MEK1* or of drug-tolerant persister cells, it is increasingly recognised that tumour cells may undergo non-genetic adaptive changes such as metabolic reprograming or reversion to a progenitor cell phenotype which result in drug resistance. It remains to be seen whether such adaptive changes will emerge in HCLc, a tumour with significantly less genomic complexity and instability [119,120,121,122,123].

## 3. Hairy Cell Variant

### 3.1. Cytogenetic and Copy Number Abnormalities

CBA showed an abnormal karyotype in 12/17 (71%) of cases, of which 5 (29%) were complex, defined as three or more chromosomal abnormalities. Recurrent aberrations included 17p abnormalities and del(18q) each in three cases. FISH analysis showed TP53 deletion or monosomy 17 in 5 of 12 (42%) cases and *ATM* deletion or monosomy 11 was detected in 2 of 9 (22.2%) cases. One case showed del(7q) by both conventional karyotype and FISH analysis [124]. Using single-nucleotide polymorphism (SNP) arrays in 15 previously untreated cases, CNAs were identified in 14 (93%) cases, with a mean of 7.9 abnormalities per case. Although the data are limited, combined CBA and SNP results suggest a greater degree of genomic complexity in HCLv than HCLc. Gains on chromosome 5 were identified in 5 cases and deletions of 17p and 7q in five and three cases, respectively [47]. Copy number analysis of regions covered in a targeted sequencing study identified 7q deletions and also recurrent 3p deletions which included a critical tumour suppressor locus encoding *VHL*, *SETD2*, *BAP1*, and *PBRM1* [11].

#### 3.1.1. Recurring Mutations

Information on the genomic landscape of HCLv is also based on limited data—namely, WGS in 7 cases published in abstract from only [125], WES in 7 cases [43] targeted sequencing using a cancer gene panel in 12 cases [11,42], and targeted sequencing of *MAP2K1* in 25 cases [42,124,126] and of *TP53* in 30 cases [127]. No case had the *BRAF-V600E* mutation. Recurring mutations have been found in *TP53*, among cases with a 17p deletion [47], in *MAP2K1*, *U2AF1*, and *KDM6A*, as discussed below, and in *ARID1A* and *CREBBP*, while single mutations were identified in *CCND3*, in genes involved in transcriptional regulation (*CEBPA*, *DDX3X*, and *PBRM1*) and chromatin remodelling (*KMT2C* and *KDM5C*).

MAP2K1

Waterfall first identified *MAP2K1* mutations in HCLv in 10/24 (41%) cases. The mutations mapped predominantly to the regions encoding the negative regulatory region and catalytic core (Figure 5) and are functionally active, increasing the basal enzymatic activity of MEK, encoded by *MAP2K1* [128]. The only non-missense mutation was a 48 bp in-frame deletion (amino acids 42 through 57) that almost entirely removed the autoinhibitory helix A13. Subsequent studies have confirmed the finding of recurring *MAP2K1* mutations but at a varying and predominantly lower incidence, with 2/4, 3/8, 2/11, and 1/14 cases and an overall incidence of 8/37 (22%) [11,42,124,126].

U2AF1

*U2AF1* encodes the U2AF1 protein which heterodimerises with U2AF2, to form a key component of the splicesome. Among haematological malignancies, *U2AF1* and *U2AF2* mutations are largely restricted to myeloid neoplasms, especially high-risk myelodysplastic syndromes and acute myeloid leukaemia in which *U2AF1* mutants may alter the differential splicing of many genes that affect various biological pathways, including DNA damage response (*ATR* and *FANCA*) and epigenetic regulation (*H2AFY*, *ASXL1*, *BCOR*, and *DNMT3B*). Subclonal hotspot p.Ser34Phe *U2AF1* mutations were identified in 2/7 cases of HCLv [43] and in one case which underwent a high-grade transformation in a lymph node 7 years after presentation, the mutation was found in both pre- and post-transformation samples [129]. The biological significance of *U2AF1* mutations in HCLv is unknown, but they are potential targets for splicing inhibitors.

KDM6A

*KDM6A* encodes a lysine demethylase protein that removes di- and tri-methyl groups from lysine 27 of Histone 3 (H3K27). Potentially deleterious mutations resulting in the loss of the highly conserved C-terminal region of *KDM6A*, essential for its demethylase activity, were identified in 2/4 cases, one of which was unresponsive to first- and second-line therapies [42]. Their biological and clinical significance in HCLv is unknown but the loss of *KDM6A* activity may sensitise tumour cells to demethylating agents such as EZH2 inhibitors [130].

#### 3.1.2. Immunogenetic Features

Immunogenetic profiling in HCLv shows distinct differences in the incidence of somatic hypermutation and *IGHV* gene usage, compared with HCLc. A study of 41 patients revealed that 22% had truly unmutated *IGHV* genes, with 100% germline identity, and 5% were borderline mutated, with 97–99.9% germline identity. The most commonly used gene was *IGHV4-34*, in 17% of cases, of which 67% were truly unmutated [74]. Similar findings were reported in 26 patients, with 5/28 (18%) of rearrangements truly unmutated and 10/28 (36%) borderline mutated. *IGHV4-34* was used in 10 (36%) cases, and all were unmutated, using a 98% cut-off [75].

### 3.2. Clinical Implications

#### 3.2.1. Prognostic Significance of TP53 Aberrations

A significant association was found between 17p deletion and shorter overall survival [127]. In a recently reported phase II study of CDA plus rituximab in 20 patients, the overall CR rate was 95%, and 80% achieved bone marrow MRD negativity at 6 months. The five patients with a *TP53* mutation had a significantly shorter PFS and OS than those with wild-type *TP53* [131].

#### 3.2.2. Targeted Therapy

In contrast to HCLc, current treatments for HCLv are suboptimal, with chemoimmunotherapy remaining the preferred initial therapy [6,132]. A patient with *IGHV4-34* expressing HCLv who had relapsed with skin nodules following multiple previous treatments, including chemoimmunotherapy and allogeneic transplantation, was found to have a somatic *MAP2K1* p.K57N mutation, with a VAF of 43.26% in skin and 20.08% in blood. He received the MEK inhibitor trametinib and achieved a partial response [133]. Based on partial or complete remissions following compassionate use of MEK inhibitors in patients with either HCLc with wild-type *BRAF* or HCLv, use of the MEK inhibitor binimetinib is currently being explored in a phase II trial of both patient groups who have relapsed or refractory disease, have received at least one course of a purine analogue, and who require further treatment. Trial inclusion criteria do not include a requirement to show evidence for dysregulation of the MAPK pathway such as a *MAP2K1* mutation or increased pERK expression [134].

## 4. Splenic Diffuse Red Pulp Lymphoma

### 4.1. Cytogenetic and Copy Number Abnormalities

The largest studies in SDRPL employing CBA identified cytogenetic abnormalities in 35–57% of cases, of which 13% had a complex karyotype. The most frequent abnormality was 7q deletion in 18–25%, while recurring trisomies of chromosomes 3, 12, or 18 were also seen [7,64,135]. In contrast, copy number analysis in 16 cases using array-comparative genomic hybridisation identified aberrations in 69% of samples, including recurrent losses of 10q23, 14q31–q32, and 17p13 in three, and 9p21 in two cases. Deletion of 7q31.3–q32.3 was present in only one case, and trisomy 3 or 18 were not detected [136].

### 4.2. Genomic Mutations

As with HCLc, the data are limited and based on WES in 33 cases, targeted sequencing using a panel of 109 genes relevant to lymphomagenesis in 42 cases, and targeted sequencing of *CCND3* in 34 cases and of *BRAF*, *MAP2K1*, *MYD88*, *NOTCH1*, *NOTCH2*, *SF3B1* and *TP53* in 23–36 cases. The most frequent recurring abnormalities involved *CCND3* and *BCOR*, as described below, while single- or low-frequency recurrent mutations were found in genes encoding proteins involved in cell cycle regulation, epigenetic regulation, the RAS–MAPK pathway, NF-κB and NOTCH signalling, cytoskeleton, and cell–matrix interactions [64,135,136]. No *BRAFV600E* mutations were found, but a single *BRAF* mutation (p.G469A) was identified in a case with a non-HCLc (CD103+, CD25-, CD123-) immunophenotype, expression of an unmutated *IGHV4-34*-encoded BcR IG and a *MAP2K1* mutation, highlighting the overlap of genomic and immunogenetic features in some cases diagnosed as HCLv or SDRPL.

#### 4.2.1. CCND3 Mutations

*CCND3* located at 12p13 encodes cyclin D3, required in normal B cells for the proliferative expansion of pre-B cells and of B cells within the dark zone of germinal centres [137,138]. In addition, cyclin D3 has several non-canonical functions which include the activation or repression of transcription either directly or via the recruitment of chromatin modifiers to gene promotors [139].

*CCND3* mutations have been identified in 20–24% cases of SDRPL and almost invariably comprise missense variants in the negative regulatory proline, glutamic acid, serine, and threonine (PEST) domain, involving the amino acids T283, P284, and I290 which are part of a phosphorylation motif that regulates cyclin D3 phosphorylation and stability [64,140].

Cyclin D3 was shown to be overexpressed in >50% of tumour cells from splenectomy samples in all cases with a *CCND3* mutation and also in 19/24 cases without a *CCND3* mutation. *CCND3/IGH* translocations resulting in cyclin D3 overexpression have previously been documented in other B-cell lymphomas, but no *CCND3* translocations were detected using a *CCND3* Break Apart FISH Probe Kit [140]. Currently, neither the functional consequences of cyclin D3 overexpression in SDRPL nor the explanation for cyclin D3 overexpression in *CCND3* WT cases is understood. However, a subsequent study in cyclin D1 negative MCL expressing cyclin D3 revealed cryptic insertion of the IGK/L enhancer upstream of the *CCND3* gene that was undetectable by standard FISH probes and was associated with *CCND3* overexpression [141].

#### 4.2.2. BCOR Abnormalities

The BCL6 co-repressor (*BCOR*) gene, located at Xp11.4., encodes the widely expressed BCOR protein whose function is highly tissue specific. In germinal centres (GC), BCOR interacts with polycomb repressive complex 1 (PCR1) and BCL6, facilitating the transient repression of genes associated with DNA damage response, cell cycle checkpoint control, GC exit, and plasma cell differentiation [142].

*BCOR* mutations, found in 16% of SDRPL cases, are characterised by splicing site (1/6), nonsense (2/6), and frameshift (3/6) alterations across the coding sequence, consistent with loss of function, as seen in other lymphoid and myeloid malignancies. Overall, 4/6 mutations exhibited a high variant allele frequency. Additionally, loss of the *BCOR* locus due either to a microdeletion or loss of a whole X chromosome was found in four SDRPL female patients, with no *BCOR* mutation within the remaining allele, resulting in an overall incidence of *BCOR* abnormalities in 11/42 cases.

*BCOR* mutations have also been identified in other non-GC-derived B-cell tumours such as SMZL, MCL, prolymphocytic leukaemia, and CLL. The functional consequences of mutations in these tumours and in SDRPL are unknown [64,143].

### 4.3. Immunogenetic Features

The great majority (79–89%) of cases have been found to carry hypermutated *IGHV* genes (<100% identity), with overrepresentation of the *IGHV3-23* and *IGHV4-34* genes. Of 10 cases using *IGHV4-34*, 6 were borderline unmutated, and 4 were truly unmutated, comparable to the findings in HCLv. *IGHV1-2* usage was confined to a single case [64,135]. Broadly similar findings were reported in another study of 13 patients [136].

## 5. Conclusions and Future Studies

A major focus of this review was the key role that the discovery of the almost ubiquitous clonal *BRAFV600E* mutation has played in understanding the biology of HCLc and its importance both in differential diagnosis and as a therapeutic target. However, there remain many unanswered questions regarding the diagnosis and biology of both HCLc and, particularly, HCLv and SDRPL. Of greater clinical importance is an unmet need for potentially curative non-chemotherapeutic regimens for HCLc and more effective treatments for HCLv which additional genetic data may help to resolve.

While the finding of a *BRAFV600E* mutation in HCLc unequivocally identifies a disorder with largely uniform laboratory and clinical features, methylome and clinical course, the pathogenesis of less frequent features such as skeletal involvement, found in 3% of cases [144], and a propensity to autoimmune disease [145] remain unexplained. Additionally, there is still much to learn about the incidence, biology, and optimal management of cases with a typical HCLc phenotype that lacks the *BRAFV600E* mutation or another mechanism for *BRAF* upregulation.

It is also unlikely to be coincidental that *BRAF WT* HCLc cases display enrichment for *IGHV4-34* gene usage, frequently accompanied by activating *MAP2K1* mutations, and that these two features are also found in a subset of cases with HCLv, raising questions about the inter-relationship between these two patient groups. If *IGHV4-34*-positive, *MAP2K1*-mutated cases of HCLc and HCLv do exhibit the typical phenotypes of HCLc and HCLv, respectively, what might account for the differences between the two phenotypes?

There is also uncertainty about the relationship between HCLv and SDRPL, given their many overlapping features and the current absence of disease-defining genetic abnormalities. The absence of reports of the rare cases of SDRPL with progressive disease acquiring typical features of HCLv such as *TP53* abnormalities or prominent nucleoli would suggest they are not simply different stages of a single disease.

These uncertainties are, in large part, a consequence of the rarity of these disorders, the lack of cell lines and animal models, and the difficulty in obtaining tumour cells, especially in HCLc, where the circulating tumour cell count is usually low, bone marrow aspiration is unsuccessful, and splenectomy rarely performed [146,147]. This is reflected in the lack of genomic data on HCLc and especially HCLv and SDRPL, compared with that available in the more common B-cell tumours, such that the published genomic landscapes are unlikely to reflect the full range or true incidence of CNAs and mutations present in all three disorders.

New biological insights are likely to require studies in larger multi-institution patient cohorts, together with the application of newer technologies such as WGS, and transcriptomic and epigenetic analyses, both at the bulk and single-cell levels, comparing data from tumour cells with that from normal splenic B-cell subsets.

It is conceivable that these studies, in conjunction with a more detailed analysis of the TME, may lead to the identification of new disease subsets within or spanning the current diagnoses of *BRAF WT* HCLc, HCLv, and SDRPL, offer new insights into their cells of origin, and give rise to a more genetically based classification, offering more precise diagnostic and prognostic features and targeted therapies.

## Figures and Tables

**Figure 1 cancers-14-00697-f001:**
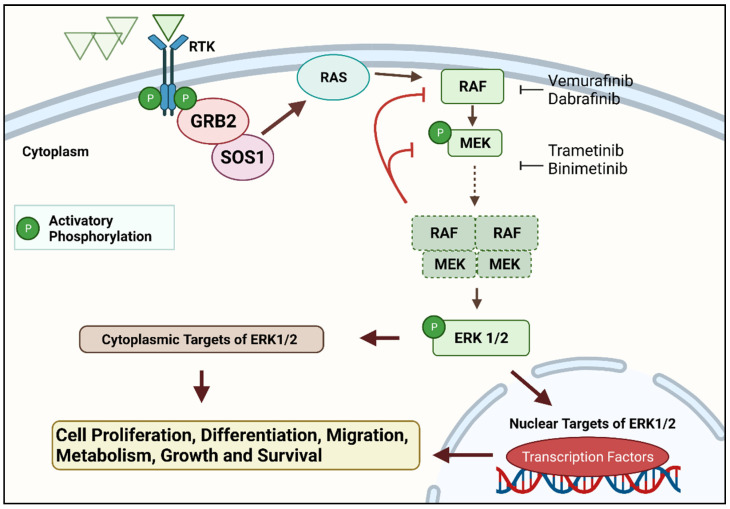
The RAS–RAF–MEK–ERK signal transduction cascade is one of four mitogen-activated protein kinase (MAPK) cascades which are activated in response to extracellular signals. RAS activation occurs within biomolecular condensates at the inner part of the cell membrane and recruits members of the RAF kinase family (A-RAF, B-RAF, and C-RAF/RAF-1) to the plasma membrane for activation. Active RAF kinases phosphorylate downstream mitogen-activated protein kinase/extracellular signal-regulated kinase ERK kinase (MEK). A transient tetramer, consisting of two RAF-MEK dimers, is formed to facilitate MEK activation by RAF. Active MEK then dually phosphorylates its only downstream targets, extracellular signal-related kinases 1 and 2 (ERK1/2). In contrast, ERK1/2 has extremely broad substrate specificity and is capable of activating both nuclear and cytosolic targets, many of which are transcription factors essential for the regulation of cell proliferation, survival, growth, metabolism, migration, and differentiation. In addition, ERKs also phosphorylate RAFs themselves at specific inhibitory amino acid residues, which releases RAF from RAS and extinguishes the signal via a negative feedback mechanism [12,13]. Created with Biorender.com (accessed on 28 December 2021).

**Figure 2 cancers-14-00697-f002:**
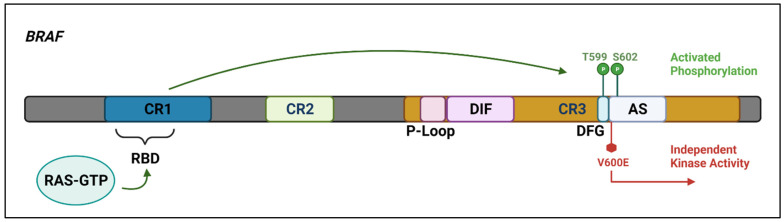
The *BRAF* protein includes three highly conserved regions—CR1 which functions as an auto-inhibitor of the *BRAF* kinase domain and contains a RAS-GTP binding domain (RBD), CR2 which acts as a flexible hinge between CR1 and CR3, and CR3, the kinase domain, which comprises multiple subregions including the P loop, the dimerisation interface (DIF), the DFG motif, and the activation segment. In the wild-type protein, inactive RAF exists in an auto-inhibited state. Under activating conditions, RAS-GTP binds to the RBD, disrupting auto-inhibition. *BRAF* is then phosphorylated at T599 and S602 within the DFG motif and activation segments, destabilising interactions with the P loop and allowing the activation segment to flip into its active conformation. The majority of *BRAF* mutants are located within either the P loop or the activation segment and adjacent DFG motif. The *BRAF-V600E* mutation occurs in the kinase activation segment, thereby inducing a change to the active conformation independently from upstream RAS activation. This results in constitutive kinase activity and aberrant signalling through the RAF–MEK–ERK pathway [14,15]. Created with Biorender.com (accessed on 28 December 2021).

**Figure 3 cancers-14-00697-f003:**
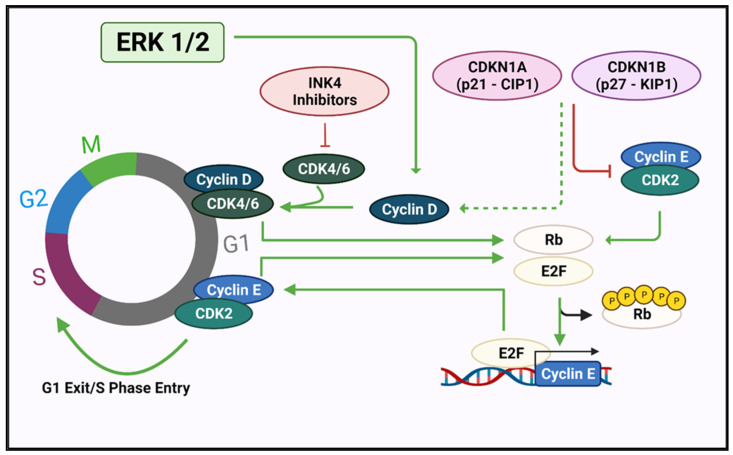
Mitogenic signals received during the G1 phase of the cell cycle, partially mediated through RAS-induced ERK signalling, upregulate the D-type cyclins D1, D2, D3, encoded by *CCND1, CCND2*, and *CCND3*, respectively. These bind and activate their catalytic partners, CDK4 or CDK6, whose activity is, in turn, negatively regulated by the INK4 family of inhibitors which include p16INK4A and p15INKB encoded by *CDKN2A* and *CDKN2B*, respectively. The formation of stable cyclin D-CDK4/6 complexes also requires the KIP/CIP proteins p21CIP1, p27KIP1, and p57KIP2, encoded by *CDKN1A*, *CDKN1B*, and *CDKN1C*, respectively, which serve as assembly factors for cyclin D-CDK4/6 but also act as inhibitors of Cdk2–cyclin E complexes required for transition into S phase [52]. Created with Biorender.com (accessed on 28 December 2021).

**Figure 4 cancers-14-00697-f004:**
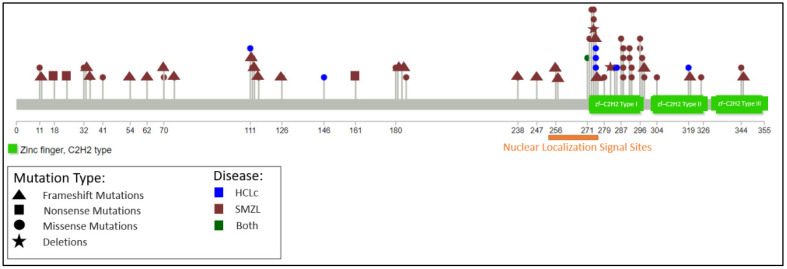
Distribution of *KLF2* mutations in HCLc and SMZL. Mutations were compiled from [42,62,64]. Diagram produced using Lollipops [65].

**Figure 5 cancers-14-00697-f005:**
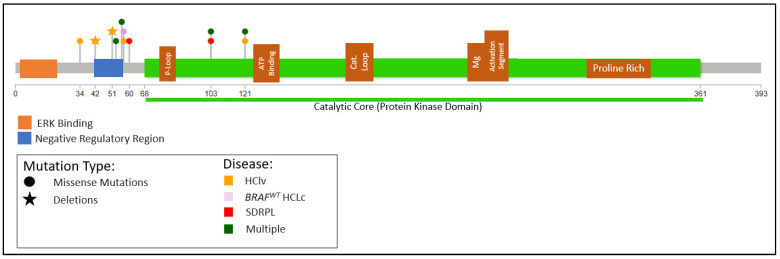
Distribution of *MAP2K1* mutations in *BRAF^WT^* HCLc, HCLv, and SDRPL. Mutations compiled from [11,42,43,62]. Diagram produced using Lollipops [65].

**Table 1 cancers-14-00697-t001:** Key Differences between HCLc, HCLv, and SDRPL.

		HCLc	HCLv	SDRPL
Demographics	Incidence	0.4/100,000	0.03/100,000	?
	Median age	55–63	70	70
	M:F ratio	3–4: 1	1–2: 1	1.6–2.4: 1
Haematology	Monocytopenia	Yes	No	No
	Nucleolus	Inconspicuous	Single prominent	Inconspicuous
Immunophenotype	Surface IgH	Usually, multiple isotypes	Usually IgG +/− other isotypes	M+/−D, M+G, G
	CD11c	Strong	Strong	Moderate
	CD25	Strong	Negative	Negative (weak in 3%)
	CD103	Strong	Moderate	Negative (weak in 33%)
	CD123	Strong	Negative	Negative (weak in 15%)
	CD27	Negative	?	Negative (positive in 20%)
	CD200	Strong	Weak or negative	Weak
Immunohistochemistry	Annexin A1	Positive	Negative	Negative
	Cyclin D1	Positive	Negative	Negative
Outcome	Need for treatment	Yes	Yes	Approx. 50%

**Table 2 cancers-14-00697-t002:** Most frequent features of HCLc, HCLv, and SDRPL.

			HCLc		HCLv	SDRPL
			HCLc *BRAF* *WT*	HCLc *BRAF* Mutated or Upregulated		
Recurring CNAs *		7q Loss		Y	Y	Y
		8q Loss (MAPK15)		Y	N	N
		17p Loss		Rare	Y	Rare
		X or Xp Loss (BCOR)		N	N	Y
		5q Gain		Y	Y	N
Genomic Mutations	MAPK Pathway	*BRAFV600E*	0%	>99%	0%	0%
		*MAP2K1*	22%	0%	22–41%	7–13%
	Cell Cycle	*CDKN1B*		16%		
		*CCND3*		0%	13%	21–24%
	Epigenetic Regulators **	*KMT2C*		15%	25%	
		*KDM6A*		0%	50%	
		*CREBBP*		5%	12–25%	
		*ARID1A*		4%	4%	9%
	Transcriptional Repressors	*BCOR*		0%	0%	16%
	NFKβ Pathway	*KLF2*		13%	0%	2%
	Spliceosome	*U2AF1*		0%	2/7	0%
		*TP53*		2%		
*IGHV* Genes	Homology	100%		<5%	18–22%	11%
		98–100%		17–20%	27–53%	14%
		<98%		80%	47–73%	86%
	Gene Usage	*IGHV3-23*		7–10%		14%
		*IGHV3-30*		7–10%		6%
		*IGHV4-34*	50%	7–10%	17–36%	22%

* % incidence of CNAs is not provided due to wide variations among studies. ** % incidence of mutations in epigenetic regulators is based on small samples.

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
