# Peer review of "The Genomics of Hairy Cell Leukaemia and Splenic Diffuse Red Pulp Lymphoma"

_cancers, 2022, doi:10.3390/cancers14030697_

Round 1

Reviewer 1 Report

In the manuscript ''The Genomics of Hairy Cell Leukemia'' the authors provide a comprehensive review of Hairy Cell Leukemia (HCL) with specific focus on variant subsets and associated genetics.  HCL is a relatively rare chronic lymphoid malignancy with distinctive clinical presentation and genetics. HCL can be subdivided into 3 sub groups, Classic HCL (HCLc), Variant HCL (HCLv) and Splenic Diffuse Red Pulp Lymphoma (SDRPL). The 3 sub types differ both in clinical presentation, age of onset, expression of immune signaling molecules and immunohistochemistry markers. These differences suggest a need to identify specific molecular targets in order to improve treatment options for HCL subtypes.  Interestingly genetic analysis has shown an almost 100% presence of mutations in the BRAF kinase in HCLc, in which an activating V600E mutation results in overactivation of the MAPK pathway. In contrast HCLv and SDRPL do not display BRAF mutations but have alterations in other MAPK pathway components and regulators. Considering the prevalence of MAPK alterations, particularly focused on BRAF, it raises the possibility of BRAF inhibitors as a potential treatment option of HCLc. Indeed the authors note that combination therapies of vemurafenib and rituximab gave favorable outcomes and increased relapse free survival times. The authors discuss summarize the various cytogenetics and mutations present in HCLv and SDRPL.

The review is comprehensive, clear, well structured and highly relevant to the field. Since HCL is a relatively rare lymphoid malignancy with distinctive presentations, this review fills a gap in the knowledge.  The cited references are current and relevant. The figures are high quality and extremely illuminating in the context of the review in summarizing the main signaling proteins and pathways involved in HCL. Overall this is an excellent body of work that successfully summarizes the field in an interesting manner that is accessibly to experts and non experts alike. The authors writing style is clear and compelling. I recommend immediate publication and commend the authors for such an excellent review. 

Author Response

We thank the reviewer for the kind comments and very positive feedback.

Reviewer 2 Report

Oscier submit a review on the genomics of hairy cell leukemia with most of the focus on BRAF mutated classic hairy cell leukemia (HCL), although the variant HCL type, splenic red pulp lymphomas and splenic marginal zone lymphoma is mentioned at places in the text.  HCL is currently a rare lymphoma of interest and the finding of BRAF alterations being targetable paves the way for improved outcomes in these patients as this is a relapsing/remitting chronic disease.  The overall review is reasonable to include as a low impact review of rare lymphomas with focus on HCL genomics with minor revisions as below:

  1. Please stick with a common theme and flow for the manuscript. In the abstract, HCL classic and variant and splenic red pulp lymphoma and splenic marginal zone lymphoma are mentioned but there is almost no mention of splenic red pulp lymphoma and no significant mention of MZL within the body of the manuscript.  If there is significant overlap between the HCL classic and variant forms and red pulp lymphoma is fine to mention aspects of those briefly with mention that the latter is a distinct entity and I would not mention MZL at all.
  2. Line 78, the first sentence in 2.1 mentions that “the causal genetic lesion in most cases was a single somatic point mutation in the DNA sequence of BRAF etc.” This is misleading as there are other alterations that clearly can be found in HCL cases.  It would be more ideal to mention BRAF V600E as the most frequent mutation etc. but it is in no way the sole driver of HCL.
  3. Figure 2 and the caption is misleading as to V600E mimicking phosphorylation. Yes it does stabilize the active form of the enzyme but by the Figure 2 it seems to suggest there is some role of phosphorylation at residual 600 which is misleading.  This mechanism can be better explained in a different fashion.
  4. I would bring section 2.1.3 to the start of section 2 (2.1) as genetic studies are mentioned (line 77) but it is not explained well what those studies are until later in the paper after the statement about BRAF in HSCs.
  5. Section 2.3…is there any germline mutations associated with HCL in these families, vis a vis a role for genetic counseling and clinic germline testing.

Author Response

We thank the reviewer for the helpful comments. In response to their corrections:

  • Please stick with a common theme and flow for the manuscript. In the abstract, HCL classic and variant and splenic red pulp lymphoma and splenic marginal zone lymphoma are mentioned but there is almost no mention of splenic red pulp lymphoma and no significant mention of MZL within the body of the manuscript.  If there is significant overlap between the HCL classic and variant forms and red pulp lymphoma is fine to mention aspects of those briefly with mention that the latter is a distinct entity and I would not mention MZL at all.

Mention of splenic marginal zone lymphoma has been removed from the abstract. While  genomic data on splenic diffuse red pulp lymphoma is limited, section 4 of the review attempts to summarise the published cytogenetic, genomic and immunogenetic data. References to SMZL and other B cell tumours in the text have been sparing and only included where we felt it was relevant to the interpretation of the data on HCL and SDRPL.

  • Line 78, the first sentence in 2.1 mentions that “the causal genetic lesion in most cases was a single somatic point mutation in the DNA sequence of BRAF etc.” This is misleading as there are other alterations that clearly can be found in HCL cases.  It would be more ideal to mention BRAFV600E as the most frequent mutation etc. but it is in no way the sole driver of HCL.

We have rephrased this sentence, omitting the word ‘causal’, as follows:

Whole exomic sequencing of a single case of HCLc lead to the discovery of a single somatic, point mutation in the DNA sequence of v-Raf murine sarcoma viral oncogene homolog B (BRAF), a kinase-encoding proto-oncogene. The same mutation was subsequently found in all 47 additional cases studied

  • Figure 2 and the caption is misleading as to V600E mimicking phosphorylation. Yes it does stabilize the active form of the enzyme but by the Figure 2 it seems to suggest there is some role of phosphorylation at residual 600 which is misleading.  This mechanism can be better explained in a different fashion.

The legend to Figure 2 has been modified to include the following sentences:

 The BRAF-V600E mutation occurs in the kinase activation segment thereby inducing a change to the active conformation independently from upstream RAS activation. This results in constitutive kinase activity and aberrant signaling through the RAF-MEK-ERK pathway.

Figure 2 has also been modified accordingly.

  • I would bring section 2.1.3 to the start of section 2 (2.1) as genetic studies are mentioned (line 77) but it is not explained well what those studies are until later in the paper after the statement about BRAF in HSCs.

We debated at what point in the review to introduce the genomic features and biological significance of BRAF V600E and while the reviewer’s comment is valid, we felt that, given the central role of this mutation in HCLc it would be advantageous, especially for readers with little prior knowledge of HCLc genomics, to discuss this early in the review. We should therefore prefer to retain the current format.  

  • Section 2.3…is there any germline mutations associated with HCL in these families, vis a visa role for genetic counseling and clinic germline testing.

Exomic sequencing of four multiplex HCL families has not identified common variants or genes among the pedigrees and as yet there is no role for genetic counseling.

Reviewer 3 Report

Major

  1. The last sentence of the simple summary and abstract are run on sentences. These should be divided into 2 or 3 sentences. 
  2. The title should reflect the manuscript. The Genomics of Hairy Cell Leukemia is the title but HCL is not the only lymphoma that is discussed. The manuscript reviews all of the lymphomas that predominantly involve the spleen then compares and contrasts them. I would recommend "The Genomics of Lymphoma with Primary Splenic Involvement" instead or something that encompasses all of the lymphomas described. 
  3. With this in mind, splenic marginal zone lymphoma is mentioned in the Abstract but then not again in the remainder of the manuscript. I can understand the desire to not include to many lymphomas but recommend that if not included it be removed from the abstract and mentioned in the introduction why it was not included. In my opinion this would better frame the manuscript. 

Minor

1. Page 1 line 38- instead of commoner would write more common

Author Response

We thank the reviewer for the helpful comments. In response to their corrections:

Major

- The last sentence of the simple summary and abstract are run on sentences. These should be divided into 2 or 3 sentences. 

The sentences in the summary and abstract have been divided into 2 sentences.

- The title should reflect the manuscript. The Genomics of Hairy Cell Leukemia is the title but HCL is not the only lymphoma that is discussed. The manuscript reviews all of the lymphomas that predominantly involve the spleen then compares and contrasts them. I would recommend "The Genomics of Lymphoma with Primary Splenic Involvement" instead or something that encompasses all of the lymphomas described. 

The title was provided by Cancers who subsequently allowed us to include a section on splenic diffuse red pulp lymphoma. Splenic marginal zone lymphoma is the subject of a separate review but we agree that the title could encompass all the lymphomas described.

- With this in mind, splenic marginal zone lymphoma is mentioned in the Abstract but then not again in the remainder of the manuscript. I can understand the desire to not include to many lymphomas but recommend that if not included it be removed from the abstract and mentioned in the introduction why it was not included. In my opinion this would better frame the manuscript. 

Mention of splenic marginal zone lymphoma has been removed from the abstract and included in the introduction

Minor

- Page 1 line 38- instead of commoner would write more common

This change has been made.